# Current Perspectives of Mitochondria in Sepsis-Induced Cardiomyopathy

**DOI:** 10.3390/ijms25094710

**Published:** 2024-04-26

**Authors:** Tatsuki Kuroshima, Satoshi Kawaguchi, Motoi Okada

**Affiliations:** Department of Emergency Medicine, Asahikawa Medical University, Asahikawa 078-8510, Japan; kurotatsu@asahikawa-med.ac.jp (T.K.); s-kawa@asahikawa-med.ac.jp (S.K.)

**Keywords:** sepsis, SICM, mitochondria, metabolic switch, cell death, mitophagy, lncRNAs, adrenergic receptor

## Abstract

Sepsis-induced cardiomyopathy (SICM) is one of the leading indicators for poor prognosis associated with sepsis. Despite its reversibility, prognosis varies widely among patients. Mitochondria play a key role in cellular energy production by generating adenosine triphosphate (ATP), which is vital for myocardial energy metabolism. Over recent years, mounting evidence suggests that severe sepsis not only triggers mitochondrial structural abnormalities such as apoptosis, incomplete autophagy, and mitophagy in cardiomyocytes but also compromises their function, leading to ATP depletion. This metabolic disruption is recognized as a significant contributor to SICM, yet effective treatment options remain elusive. Sepsis cannot be effectively treated with inotropic drugs in failing myocardium due to excessive inflammatory factors that blunt β-adrenergic receptors. This review will share the recent knowledge on myocardial cell death in sepsis and its molecular mechanisms, focusing on the role of mitochondria as an important metabolic regulator of SICM, and discuss the potential for developing therapies for sepsis-induced myocardial injury.

## 1. Introduction

Sepsis is a severe condition that can be complicated by life-threatening organ dysfunction resulting from an abnormal host response to infection. However, SICM specifically affects the heart [1,2,3,4]. The mortality of patients with SICM is 30–70%, which is 2–3-fold higher than that of patients with non-cardiac-related sepsis [5,6].

Cell death in the myocardium may result in the inability to maintain normal cardiac function because most adult cardiomyocytes are terminally differentiated and non-regenerating cells [7]. Currently, The Nomenclature Committee on Cell Death reports multiple modes of cell death [8]. Cardiomyocytes have specific cell death modality and the key cell death modalities include apoptosis, necroptosis, mitochondrial-mediated necrosis, pyroptosis, ferroptosis, and autophagic cell death [9]. The cell membrane remains intact in the case of apoptosis, ferroptosis, and autophagy, while death by necroptosis, mitochondrial-mediated necrosis, and pyroptosis leads to cell membrane disruption [9]. In sepsis, a complex condition when patient background is included, myocardial cell death is not a single event, but rather causes apoptosis, necroptosis, pyroptosis, and ferroptosis simultaneously or in succession [8].

Sepsis is characterized by the acute release of multiple inflammatory mediators, including tumor necrosis factor (TNF)-α, interleukin (IL)-6, and IL-1β, and excessive release of inflammatory mediators damages tissues and organs. There is also growing evidence linking inflammation and cell death [10,11,12]. Therefore, the pathogenesis of sepsis is intimately involved in inflammation-mediated dysregulation of cell death leading to multiple organ failure.

While evidence of pathological changes in other organs has accumulated, little has been reported on the pathology of SICM. Some animal studies have shown that while myocardial mitochondria did not exhibit morphological abnormalities, there were noticeable alterations in the surrounding endoplasmic reticulum [13]. Furthermore, accumulated lipid droplets were identified in cardiomyocytes implying that impaired transport of fatty acids to mitochondria may be a potential mechanism [14]. These findings suggest SICM is caused by functional disorders in mitochondria. Indeed, SICM is considered reversible in the clinical practice.

Recent studies suggest that the pathogenesis of SICM involves disproportionate expression of pro- and anti-inflammatory cytokines, abnormal expression of Toll-like receptors and related downstream pathways, nitric oxide (NO) release, inducible NO synthase (iNOS) contributing to metabolic disturbances and reactive oxygen species (ROS). These cause complement activation, abnormal calcium processing, downregulation of adrenergic pathways, cardiomyocyte apoptosis, autonomic nervous system dysfunction, coronary microvascular dysfunction, mitochondrial dysfunction, sarcomere and mitochondrial protein downregulation [15,16,17,18]. In the heart, the systemic inflammatory response impairs the ability of energy metabolism to produce adenosine triphosphate (ATP), resulting in cardiac dysfunction [14,19].

Seen from a different perspective, SICM may be regarded as a metabolic disorder. Myocardial energy metabolism encompasses a diverse array of pathways, with mitochondria playing a central role as crucial organelles involved in metabolism. This review aimed to present recent findings in SICM, with a particular focus on abnormalities in mitochondrial function.

## 2. Methodology Literature Search

Literature searches were conducted using PubMed and major journals of clinical and basic science. Searches focused on mitochondrial roles in sepsis-induced cardiomyopathy. PubMed searches for variations of the terms “mitochondria and sepsis”, “pathophysiology and sepsis induced cardiomyopathy”, and relevant keywords were used to identify relevant papers which are available in English.

## 3. The Role of Mitochondria as a Powerhouse in Heart

The mitochondria are main source of cardiac energy. Mitochondria produce 95% of cellular ATP [20]. Generally, 60–90% of ATP production is dependent on fatty acids oxidation (FAO), and the remainder originates from the oxidation of glucose and lactic acid, ketone bodies, and amino acids in the heart [21]. The mitochondria are divided into two sections by the inner and outer membranes. The innermost space is called the matrix, and the interstitial space between the inner and outer membranes is called the intermembrane space. Electron transfer in respiratory chain complexes I–IV pumps protons from the mitochondrial matrix into the intermembrane space, creating an electrochemical gradient (∆pH) of protons across the mitochondrial inner membrane. The ∆pH and mitochondrial membrane potential (∆ψm) become the proton motive force, and ATP production is carried out by ATP synthase, called oxidative phosphorylation (OXPHOS) [22,23].

The mitochondria are important for myocardial contraction. In 2010, the previously unidentified mitochondrial Ca^2+^ uptake protein mitochondrial calcium uptake 1 was identified. Since then, it has been understood there is a coupling between Ca^2+^ signaling and mitochondrial Ca^2+^ signaling and mitochondrial metabolism. Ca^2+^ is involved in cardiomyocyte excitation and contraction. Ca^2+^ plays a crucial role in the excitation and contraction of cardiomyocytes. It is responsible for orchestrating the excitation–contraction coupling process within cardiomyocytes, where Ca^2+^ facilitates muscle contraction. Additionally, mitochondrial Ca^2+^ levels regulate the excitatory and contractile functions of cardiomyocytes. Overall, Ca^2+^ is central to maintaining the intricate balance in the excitatory-contraction relationship of cardiomyocytes [24,25].

## 4. Pathophysiology of SICM

The sequence of events leading to SICM involves several circulating factors. These extracellular mediators include both pathogen-associated molecular patterns (PAMPs), such as lipopolysaccharides (LPSs), and host-produced damage-associated molecular patterns (DAMPs). LPSs are exclusively found in the outermost membrane of Gram-negative bacteria and bind to Toll-like receptor 4 to produce inflammatory mediators. DAMPs also include cytokines (IL-1β, TNF, IL-6) [26,27,28], iNOS [29,30,31], ROS [32,33], heat shock proteins, high mobility group box 1, histones, activated complement components, and mitochondrial DNA. Preclinical studies have shown that these molecules directly or indirectly suppress cardiac function. However, no correlation has been found between cytokines measured in septic patients and myocardial dysfunction. This suggests that the ultimate effects of these circulating factors on the heart are likely due to the interaction of a wide range of signaling pathways rather than a single factor. Myocardium from septic animals had normal high-energy phosphate levels and no evidence of cellular hypoxia [34,35,36]. Other abnormalities which proposed to contribute to the SICM etiology include altered calcium transport [37], altered metabolism [38], impaired myocardial electrical conduction [39,40], and adrenergic signaling abnormalities [41,42]. However, none of these mechanisms alone can satisfactorily explain why septic cardiomyocytes, once in a state of functional arrest, rapidly regain function [34]. The mitochondria are strongly associated with the above-mentioned SICM factors, and examining the role of this organelle is worthwhile for discussion.

### 4.1. Mitochondrial Structural Abnormalities

The role of nuclear fission and fusion was studied in experimental sepsis, but few researches have focused on cardiac mitochondria. Electron microscopy of rat hearts 24 h after endotoxin administration revealed compatible morphological changes with the processes of nuclear fission and fusion [43]. An imbalance in a severe cecal ligation and puncture (CLP) model of mouse sepsis between mitochondrial fission and fusion with dynamin-related protein 1 (Drp1) activation and optic atrophy 1 (OPA1) downregulation was demonstrated in association with mitochondrial structural abnormalities, mitochondrial dysfunction, and reduced cardiac contractility [35]. The heart of LPS-treated mice also showed Drp1 activation, with decreased mitochondrial size, increased fragmentation, and abnormal morphology and function [36]. In contrast, OPA1 expression was mildly increased by sublethal LPS doses [44]. These conflicting results may reflect the extent of myocardial injury and the phase of sepsis.

### 4.2. Autophagy and Mitophagy in SICM

Autophagy is one of the mechanisms involved in cellular homeostasis and cell death. Autophagy is performed by a process in which damaged proteins and organelles are placed in double-membrane vesicles (autophagosomes) and sent to lysosomes for degradation.

Mitophagy is a type of selective autophagy that aims to remove dysfunctional mitochondria and induce cell death pathways before mitochondrial outer membrane permeability is compromised. Autophagy is activated in the heart during sepsis, but whether this is protective or detrimental is unknown. However, activation of autophagy and mitochondrial depletion in the heart have been observed in a number of rodent CLP and LPS models [36,39,45].

Autophagy activation in the heart has been confirmed within 4 h in a mouse CLP model. The co-localization of autophagosomes and lysosomes is reduced in septic animals despite the increased autophagic vacuoles, suggesting impaired autophagosome interactions and decreased degradation. This incomplete autophagy is associated with cardiac dysfunction, ATP depletion, apoptosis, and necrosis, which are all restored by rapamycin, which stimulates complete autophagy. Similar findings were observed in vitro, where autophagy induction protected cardiomyocytes from LPS-induced cell death, while autophagy inhibition had the opposite result [40].

LPS stimulation was also found to induce not only myocardial autophagy but also more selective mitophagy processes [36]. This removal of damaged mitochondria via activation of mitophagy may facilitate the resolution of cardiac and mitochondrial dysfunction. Defects in sestrin 2, a protein involved in mitochondrial priming in autophagy, were associated with increased mortality. In contrast, loss of Rubicon, a key protein that negatively regulates autophagosome maturation, the Beclin-1 binding protein, enhanced autophagy flux and cardiac function [40,46].

Cardiac autophagy was upregulated in LPS-treated mice, suggesting that this phenomenon is associated with a detrimental increase in oxidative stress. Cardiac-specific overexpression of the endogenous mitochondrial antioxidant thioredoxin-1 also improved prognosis in CLP mice, but this was associated with stimulation rather than inhibition of autophagy [45]. Pharmacological inhibition of autophagy reversed contractility of neonatal cardiomyocytes but increased apoptosis and mitochondrial dysfunction in both neonatal cardiomyocytes and HL-1 cells [43,47]. Cells replace damaged mitochondria that are removed by mitophagy through mitochondrial biogenesis and interaction between peroxisome proliferator-activated receptors (PPARs) and PPAR-γ co-activator (PGC)-1α and -β. The phenomenon activates multiple transcription factors, including nuclear respiratory factors 1 and 2, to promote the expression of mitochondrial transcription factor A (T-fam). This coordinated signaling cascade increases mitochondrial DNA copy number and mitochondrial mass. Recently, mitochondrial biogenesis in cardiomyocytes has been reported in several experimental sepsis models, which still remains elusive.

### 4.3. Calcium Transport-Induced Membrane Potential in SICM

Calcium uptake into cardiac mitochondria is facilitated by the presence of microdomains between the sarcoplasmic reticulum (SR), the intracellular calcium store, and the mitochondria. These microdomains bring calcium release sites and calcium uptake sites into proximity and are maintained by tethering proteins, such as mitofusin, which are also involved in the fission or fusion process. This system allows mitochondrial calcium, the main determinant of ATP supply, to be synchronized with the ATP demand generated by the excitation–contraction coupling process [48]. Calcium transport to mitochondria is also essential for maintaining proper mitochondrial antioxidant capacity and mitigating the increase in ROS formation caused by increased ATP synthesis [49]. Calcium loading in mitochondria has detrimental effects and is a major determinant of mPTP opening, especially in the presence of oxidative stress [50]. Thus, it is easy to understand the involvement of calcium transport defects in mitochondria with the pathophysiology of many diseases, including heart failure [51]. Abnormal intracellular calcium homeostasis has been studied in the septic heart. In most septic models, cytosolic calcium transients (i.e., the difference between systolic and diastolic calcium concentration) were decreased, which was associated with increased diastolic cytosolic calcium and decreased SR calcium content [52]. These findings may be attributed to the dysfunction of SR calcium transporters, specifically the “leaky” ryanodine receptor (RyR) and SERCA, which cause increased calcium release and decreased reuptake, respectively. Changes in intracellular calcium concentrations are exacerbated by myofibrillar desensitization to calcium [53] or by altered expression of calcium-processing proteins [54]. Despite a large number of studies suggesting a central role for intracellular calcium imbalance in SICM, few studies have clearly evaluated the role of calcium in myocardial mitochondria. Myocardial mitochondrial calcium concentrations were elevated in endotoxin-treated rats in association with abnormal mitochondrial respiration, membrane potential, and myocardial dysfunction [55,56]. In cardiomyocytes exposed to endotoxin for 1 h, mitochondrial calcium concentration increased in a dose-dependent manner of endotoxin, inducing cell death. The results that treatment of these cardiomyocytes with dantrolene ameliorated the excess calcium concentration in mitochondria and inhibited cell death suggest that SICM induces a defect related to calcium transport in cardiac mitochondria [57].

Experiments of electrical pacing in automobile cardiomyocytes isolated from septic rats evaluated changes in mitochondrial calcium and found that the rate of calcium increase was lower than in control cardiomyocytes and was associated with other signs of mitochondrial dysfunction. Interestingly, the potential contact area between mitochondria and SR was reduced, and the distance between these organelles was increased by electron microscopy [45]. These structural abnormalities in the organelle may cause dysfunction of the mitochondrial-SR microdomain, leading to a reduced rate of calcium uptake. On the other hand, these findings may be an adaptive and protective response to prevent mitochondrial calcium overload at the expense of reduced energy efficiency. Myocardial calcium homeostasis has not been evaluated in human patients with sepsis. However, recent large observational studies have reported that chronic use of calcium antagonists up to hospitalization is associated with reduced mortality from sepsis [58,59], and other experimental studies have revealed specific beneficial effects of calcium antagonists on cardiac function [60,61].

Here, we show a schematic figure of mechanisms of mitochondrial dysfunction in SICM (Figure 1).

### 4.4. Mitochondrial Metabolic Abnormality in SICM

Lipid metabolism is the primary source of ATP production in the heart. Most patients are starved during sepsis and require energy from lipid mobilization and oxidation [62]. Along with increased catabolism, increased lipolysis in adipose tissue, the largest source of energy in the body, is compensated for by the conversion of triglycerides (TGs) to fatty acids (FAs) and glycerol, which are released into the bloodstream [63]. The released FAs enter the peripheral organs and are oxidized to produce ATP. However, the inflammatory response of sepsis is differentially regulated by various genes related to lipid metabolism. For example, LPS decreases PPAR and PGC1, which regulate oxidative pathways [64], and cluster of differentiation 36 (CD36) and carnitine palmitoyltransferase I (CPT1) dysregulation induce FA transport defects, leading to FAO failure [14,39,65]. In particular, CPT1 reduction prevents FA from entering the mitochondria. This leads to intracellular accumulation of lipids and organ dysfunction, as well as “lipotoxicity”. The association between myocardial lipid accumulation and heart failure in humans has long been reported. Studies reported intramyocardial triacylglycerol accumulation and its transcriptional profile in patients with heart failure, leading to lipotoxicity and contractile dysfunction, as well as documented lipid accumulation in heart samples from patients with severe chronic heart failure undergoing heart transplantation [66,67]. Lipid accumulation has been associated with cell apoptosis mechanisms [68,69], ROS production [70,71], endoplasmic reticulum stress, and cell death [72] and may itself lead to heart failure (Figure 2).

### 4.5. Pyroptosis in SICM

Pyroptosis is a type of cell death mediated by inflammations. The cell death mechanism can not only defend against microbial infection but also induce excessive host-dependent inflammatory response [73,74]. Cardiomyocytes’ pyroptosis and inflammation contributes to SICM.

LPS activates NLR family pyrin domain containing 3 (NLPR3) in myocardium of septic mice [75]. NLRP3 inflammasome is a macromolecular protein complex that senses the damage and activates inflammatory response. NLPR3 activates caspase 1 followed by IL-1β and induces pyroptosis. Actually, inhibitors of NLPR3 reduces inflammation and preserve cardiac function [76,77].

Gasdermin D (GSDMD) is also involved with pyroptosis [78]. GSDMD is expressed in a variety of cell types and works as a specific substrate of inflammatory caspases. GSDMD not only increases inflammatory factors via activation of NLPR3 but also causes decrease in ATP production by mitochondrial injury. Shanshan et al. reported that LPS increased GSDMD expression and triggered SICM, while *Gsdmd*^−/−^ mice attenuated cardiac dysfunction in sepsis [79]. In addition, Zhang et al. identified inhibition of caspase 11, and GSDMD improved SICM by attenuating cardiomyocytes’ pyroptosis [80].

### 4.6. Ferroptosis in SICM

Ferroptosis is a nonapoptotic cell death program distinct from apoptosis or necrosis. This mechanism is featured by iron-dependent lipid peroxidation and involved in various pathological processes, including neurotoxicity, cardiovascular diseases, cancer, and sepsis [81,82]. Ferroptosis is involved with depletion of glutathione (GSH) [83,84] and inactivation of glutathione peroxidase 4 (GPX4) [83,84], which results in metabolic imbalances of iron and lipids. Activation of ferroptosis causes mitochondrial swelling and mitochondrial membrane potential collapse through mitochondrial permeability transition pore (MPTP) opening, ultimately leading to cell death [84], while mitochondrial dysfunction impairs iron metabolism, which results in excessive free iron accumulation in mitochondria and lipid peroxidation of their membranes. Lipid peroxidation accumulation in the mitochondria induces cysteine deprivation and promotes glutaminolysis, thereby activating the tricarboxylic acid cycle. This activity increases mitochondrial hyperpolarization and ROS production, resulting in induction of lipid peroxidation and ferroptosis. Therefore, evidence has identified varied interplay between ferroptosis and mitochondrial dysfunction. Recently, the development of sepsis has been proposed to involve ferroptosis, and many basic studies focus on inhibition of ferroptosis to reduce organ damage including SICM [82,85].

## 5. Potential Therapeutic Interventions: Future Perspective

Mitochondria have many important roles for cardiac metabolism, and mitochondrial dysfunction in sepsis leads to cardiac dysfunction. Therefore, mitochondria-targeted metabolic resuscitation may become a promising strategy against SICM.

Here, we discuss three potential approaches of mitochondrial resuscitation in SICM: (1) antioxidative therapy to attenuate oxidative damage in mitochondria, (2) metabolic modulation of ATP production in mitochondria, (3) regulation of noncoding RNA associated with mitochondrial metabolism.

### 5.1. Therapeutic Strategies for Mitochondria with Antioxidants

Many attempts at mitochondrial therapy with antioxidants have been made in clinical studies, but with limited success [86]. Recently, a multicenter open-label RCT was conducted in severe septic patients with combination therapy of vitamin C, hydrocortisone, and thiamine, which have not shown any significant differences in survival time or free duration of vasopressor administration in comparison to those with only hydrocortisone therapy [87]. This can be predominantly attributed to conventional antioxidants’ nonspecific intracellular localization and inability to transport across multiple biological barriers and exert therapeutic effects in the target cells, the mitochondria [88]. For these reasons, the next challenge was to create antioxidants that had been chemically modified to accumulate selectively in mitochondria. This approach can be broadly divided into two approaches as follows: (1) the mitochondrial matrix has a negative potential in comparison to the cytoplasm and extracellular space, and large diameter cations are selectively sequestered within the mitochondrial matrix; and (2) the use of lipophilic side chains facilitates the movement of molecules through the mitochondrial membrane. Mitoquinone (ubiquinone attached to a triphenyl phosphonium cation), mitotempol (tempol attached to a triphenyl phosphonium cation; a similar structured related molecule is midtempo), SKQ1 (plastoquinone decyltriphenyl phosphonium), and SS-31, a small mitochondrially targeted peptide drug, have been developed using these methods [89]. Damon et al. found that mitoquinone suppresses ROS production and maintains mitochondrial membrane potential in an in vitro endothelial cell model of sepsis and that administration to septic animals reduces liver and kidney damage [90]. Mitoquinone has also been shown in animal models of sepsis to reduce cardiac mitochondrial and contractile dysfunction [91]. Furthermore, Mito-TEMPO, a mitochondria-targeted antioxidant, reduces renal impairment [92] or liver injury [93] in septic animal models, which might have a potential to attenuate SICM. SS-31 therapy effectively protected cardiac function from septic injury by suppressing inflammatory mediators and maintaining mitochondrial membrane potential [94].

The preclinical data of therapeutic approaches targeting mitochondria have accumulated. However, there is still no evidence in septic patients with SICM. Clinical trials need to be conducted to investigate whether mitochondrial-targeted antioxidants make profits without side effects against SICM.

### 5.2. Therapeutic Strategies for Mitochondrial Metabolism in SICM

Therapeutic strategies targeting different SICM mechanisms, especially mitochondrial targeting, may be necessary to achieve a more effective prognosis in practice. However, real-world clinical SICM therapy application has not yet been realized, probably due to the difficulty in fully modeling the characteristics of sepsis due to the complexity of background factors and mechanisms of metabolic abnormalities in SICM. Here, we discuss the therapeutic strategies for SICM with potential clinical applications.

Mitochondrial dysfunction impairs cardiac metabolism in sepsis. As described before, inflammation decreases the expression of CD36 and CPT1, which are important lipid transporters into mitochondria, leading to lipid accumulation. Recently, we reported that beta-3 adrenergic receptor antagonist improves lipid accumulation in SICM [14]. In the study, we found a gap in the recovery time of CD36 and CPT1 expression in LPS-induced septic hearts, suggesting that this time gap may lead to an imbalanced FA supply and demand between the cytoplasm and mitochondria, causing lipid accumulation in the cytoplasm and myocardial tissue of endotoxin model mice. Myocardial lipid droplets (LDs) were observed in histological findings in the myocardium of LPS mice using Oil Red O-stained light microscopy, whereas few LDs were observed in the myocardial mice in the normal control group. SEM with osmium immersion revealed three-dimensional intracellular ultrastructure: myocardial mitochondria in the normal control group of mice were round, plate-like clusters densely packed in the matrix space. The matrix space was sparse in LPS mice, and LDs were found outside the mitochondrial structure. These findings were consistent with light microscopic examination of tissue stained with Oil Red O (Figure 3). Interestingly, this study observed many LDs around the myocardial mitochondria 12 h after LPS injection, indicating cardiac dysfunction, but not at 24 h. This may suggest a strong relationship between lipid dysregulation and reversible cardiac function. Furthermore, in this study, we demonstrated the effects of β3ARs on mitochondrial metabolism in SICM and revealed that β3AR blockade improves cardiac dysfunction and mortality in LPS-induced lipid accumulation. β3ARs are mostly found in adipocytes and are associated with lipolysis, while a few β3ARs are present in the heart. Notably, β3ARs are increased in ischemic heart failure [95], and β3AR stimulation improves cardiac function after myocardial infarction via NOS regulation [96,97]. However, contrary to our expectations, β3ARs blockade significantly improved cardiac ATP and mortality in LPS-induced sepsis mice. This study demonstrated that (1) β3AR is increased in SICM; (2) β3AR blockade maintains cardiac ATP by improving FAO; (3) β3AR blockade prevents myocardial lipid accumulation; and (4) β3AR regulates excess NO, which induces mitochondrial dysfunction. On the other hand, many investigators argue that β3AR activation is useful in improving cardiovascular pathology [98]. Identifying the phase of the “metabolic switch” is important because the pathogenesis of sepsis involves changes in immunity and metabolism over time. Cardiac mitochondrial metabolism in sepsis is complex, but an approach to mitochondrial metabolism could provide a new therapeutic strategy for SICM [14,99].

### 5.3. Effects of Noncoding RNA Regulation on Mitochondrial Dysfunction in SICM

Recently, the involvement of noncoding RNA (ncRNAs), especially long noncoding RNAs (lncRNAs), in the mitochondria has attracted much attention. Cheng Xing Peng et al. studied the regulatory role of myocardial-infarction-associated transcript (MIAT) in SICM and revealed that knockdown MIAT significantly suppressed mitochondrial ROS production in LPS-treated HL-1 cells. This result suggested that MIAT exacerbates myocardial injury by promoting oxidative stress, and MIAT in SICM directly acts on miR-330-5p to upregulate the TRAF6/NF-kB pathway, thereby promoting inflammation and oxidative stress [100]. Ying Han et al. demonstrated that receptor-activated modifying protein (RMRP) inhibits the post-transcriptional modulatory effect of miR-1-5p on heat shock protein 70 protein 4 (HSPA4) in LPS-induced mitochondrial damage, and RMRP overexpression reduces mitochondrial membrane potential (MMP), intracellular ROS levels, cytoplasmic cytochrome c, caspase-9, and caspase-3, and significantly suppresses cardiomyocyte apoptosis [101]. Bin Shan et al. revealed that H19 modulates MMP by regulating the miR-93-5p/SORBS2 pathway and overexpressing H19 in LPS-induced cardiomyocytes markedly reduced inflammatory factors, including TNF-a, IL-1b, and IL-6, indicating that overexpression of H19 is associated with inflammation [102]. Studies on the involvement of lncRNAs in regulating mitochondrial metabolism in SICM are ongoing, and Dongshi et al. revealed that suppressing LPS-induced X-inactive specific transcript RNA (Xist) expression increases ATP production and reduces LPS-induced myocardial damage [103]. Recent studies supporting the implication of ncRNAs targeting mitochondria in SICM are summarized in Table 1.

To date, researchers at Zhejiang University have identified 471 upregulated lncRNAs and 804 downregulated lncRNAs in myocardium of septic mice using gene chip hybridization technology. Finally, this group found that partial lncRNAs are mainly enriched in inflammation, immunity, energy metabolism, and cell death and predicted that certain lncRNAs may be involved with mitochondrial dysfunction [104]. Whether other lncRNAs participate in the regulation of mitochondrial function and the specific regulatory mechanisms of the involved lncRNAs in SICM remains unclear, although the aforementioned lncRNAs were implicated in SICM by regulating mitochondrial function and apoptosis. The involvement of lncRNAs in mitochondrial metabolic disturbances in SICM awaits further elucidation.

### 5.4. Blockade of β Adrenergic Receptor in Sepsis

Parker et al. first proposed in a 1984 study that patients with sepsis exhibited intrinsic myocardial dysfunction, with an increased volume and a decreased ejection fraction, which was reversible. Although cardiovascular abnormalities in sepsis were known since the 1950s [108,109], the concept of reversible cardiac dysfunction in sepsis is novel. To our best knowledge, the mechanism of cardiac dysfunction in sepsis might be different from that of myocardial ischemia. For example, a clinical study demonstrated that coronary flow in patients with septic shock is similar to or higher than that in patients without septic shock [110]. A high level of circulating catecholamines downregulates the response of the adrenergic receptor (AR) in cardiomyocytes in the state of SICM, thereby blunting the contractile response of cardiomyocytes to catecholamines [111,112,113]. This means that vasopressors for maintaining blood pressure do not work as SICM treatment.

Recent studies have shown that βAR blockers have received unprecedented attention as a septic shock treatment. Catecholamine overstimulation in septic shock may play an important role in SICM development through calcium overload and βAR signaling and βARs downregulation [113,114,115]. In addition, β signaling-mediated calcium loading results in opening of mitochondrial permeability transition pore (MPTP) and increased ROS production that triggers mitochondrial dysfunction and apoptosis [116]. Accumulated evidence has reported that βAR blockade for sepsis reduces arrhythmogenesis, prevents SICM, and even improves mortality [117,118,119]. Notably, βAR blockers are an attractive strategy against sepsis and inhibit the excessive release of catecholamines and mitochondrial dysfunction, preventing the hypermetabolic state in sepsis. Furthermore, β1AR blockade decreases heart rate and prolongs the filling time of the left ventricle, thereby increasing single-beat output and maintaining cardiac output.

### 5.5. Other Approaches

Having discussed mitochondrial treatment in the myocardium, we discuss approaches to treating mitochondrial dysfunction itself, not just in the myocardium.

One approach to treat mitochondrial dysfunction is to activate cellular programs that replenish damaged proteins and promote mitochondrial biogenesis. PGC1α, a transcriptional activator that interacts with PPAR gamma, is recognized as an important regulator that determines the cellular production of mtDNA-dependent mitochondrial proteins. Various agents were identified as agonists of the PPAR γ, including pioglitazone and rosiglitazone. Studies revealed that these agents can potently induce mitochondrial biosynthesis in animals and humans and prevent cell dysfunction and cell death in response to mitochondria-damaging stimuli [120].

Another way to promote mitochondrial biosynthesis is with sirtuin activators. Resveratrol is a potent sirtuin-1 (Sirt1) activator, that promotes mitochondrial biogenesis, enhances oxidative metabolic capacity, and shows efficacy in animal models of cardiovascular disease, metabolic syndromes, and muscle disease. Additionally, the administration of human recombinant transcription Factor A, a mitochondrial protein with mitochondrial target sequences modulates mtDNA replication, and rhTFAM reduces mortality in animal models of sepsis [121,122,123].

Another approach is to regulate pyroptosis in cardiomyocytes. GSDMD-dependent pyroptosis is considered to be one of the causes of SICM. Irisin, which is a product of fibronectin type III domain-containing protein 5, protects heart from inflammation via upregulation of mitochondrial ubiquitin ligase (MITOL). Irisin attenuates cardiac dysfunction and cardiac damage in sepsis by activating MITOL and inhibiting GSDMD-dependent pyroptosis in vivo and in vitro [124].

In addition, inhibition of ferroptosis may be also effective. Ferroptosis is one of the critical mechanisms contributing to SICM. Ferrostatin-1 (Fer-1) is a suppressor of ferroptosis by inhibition of lipid peroxidation. Fer-1 improves survival rate and cardiac dysfunction of septic mice by attenuating ROS damage in mitochondrial [125]. Moreover, Resveratrol, a Sirt1 activator, also inhibits ferroptosis. Zeng et al. reported the SICM in CLP rats was induced by mitochondrial dysfunction, increased lipid peroxidation, and reduction in Sirt1/NF-E2-related factor 2 (Nirf2) axis which induces ferroptosis. Intriguingly, high dose of Resveratrol attenuates ferroptosis via upregulation of Sirt1/Nrf2 signaling pathways and improves SICM [126].

Another method to restore mitochondrial function is to directly transplant high-quality normal mitochondria into target tissues. In vivo, mitochondria injected or perfused into cardiac tissue are rapidly taken up by cardiac cells. Furthermore, these studies reveal that mitochondrial transplantation increases myocardial energy production and decreases cell death [127,128].

## 6. Conclusions

SICM represents a significant manifestation of sepsis linked with a poor prognosis, and this review delineates the involvement of mitochondrial dysfunction in the SICM pathogenesis. The mechanisms leading to mitochondrial dysfunction range from mitochondrial structural disorders to abnormal mitochondrial metabolism, and the temporal sequence of these mechanisms are not clearly understood. Evidence has accumulated from numerous laboratory experiments and clinical data that ATP depletion due to metabolic abnormalities is a contributing factor in SICM, which may involve mechanisms that differ from known metabolic pathways in SICM. Furthermore, even the extent to which each mechanism contributes to mitochondrial dysfunction is currently unknown. Mitochondrial metabolism, which is activated in sepsis, can be protective if moderate and progress detrimentally if excessive. This means that SICM may represent an adaptive protective mechanism with a trade-off between short-term organ function and long-term tissue viability. On the other hand, persistent mitochondrial abnormalities that do not recover may be involved in the transition from adaptive to maladaptive organ dysfunction. Therefore, the therapeutic strategy for SICM would be to ensure that the protective effect is not compromised and to provide timely therapeutic intervention.

We need to establish a definition and diagnostic criteria for SICM. We believe that increased knowledge of the molecular mechanisms associated with myocardial cell death and its mitochondrial dysfunction in sepsis will facilitate the development of targeted therapies for SICM.

## Figures and Tables

**Figure 1 ijms-25-04710-f001:**
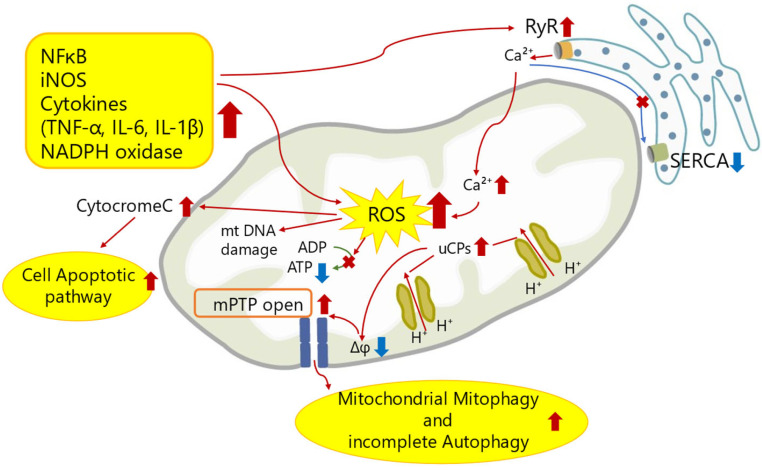
A schematic figure of mechanisms of mitochondrial dysfunction in SICM. The inflammatory response of sepsis attenuates Ca^2+^ uptake in the sarcoplasmic reticulum by upregulation of RyR and downregulation of SERCA. This abnormal Ca^2+^ flux increases Ca^2+^ concentration in mitochondria, followed by abnormal mitochondrial membrane potential and promoting mPTP opening. Theses leads to excess ROS generation, mitochondrial DNA damage, mitochondrial mitophagy, incomplete autophagy, and cell apoptotic pathway. Red arrows indicate increased expression or activity. Blue arrows indicate decreased expression or activity. A cross mark indicates no function. Abbreviations are described at the end of this manuscript.

**Figure 2 ijms-25-04710-f002:**
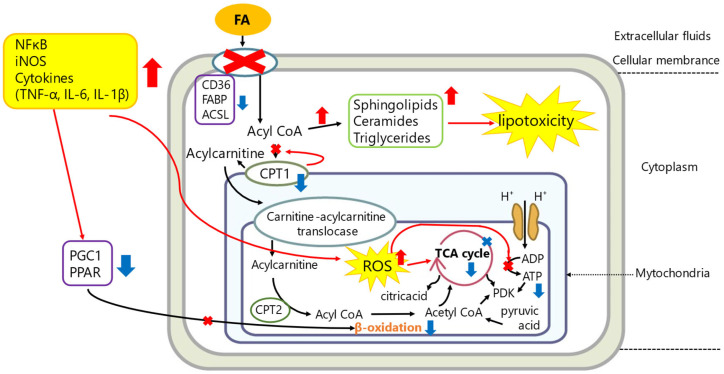
A schematic figure of metabolic dysregulation in SICM. ATP synthesis in the heart is mainly dependent on fatty acid oxidation (FAO). The inflammatory response in sepsis is induced by many inflammatory mediators such as NFκB, iNOS, cytokines, and ROS, which dysregulate various genes related to FAO. Inflammation reduces CD36, FABP, and ACLS, which are cell surface transporters for fatty acids (FAs) into cells, and also reduces CPT1, which is an important enzyme for intracellular FAs to enter mitochondria. In addition, inflammatory mediators also decrease PPARs and PGC1α essential for β-oxidation. As a result, compromised ATP production induces intracellular lipid accumulation, called lipotoxicity, leading to cell death. Red arrows indicate increased expression or production. Blue arrows indicate decreased expression or production. Cross marks indicate no function. Abbreviations are described at the end of this manuscript.

**Figure 3 ijms-25-04710-f003:**
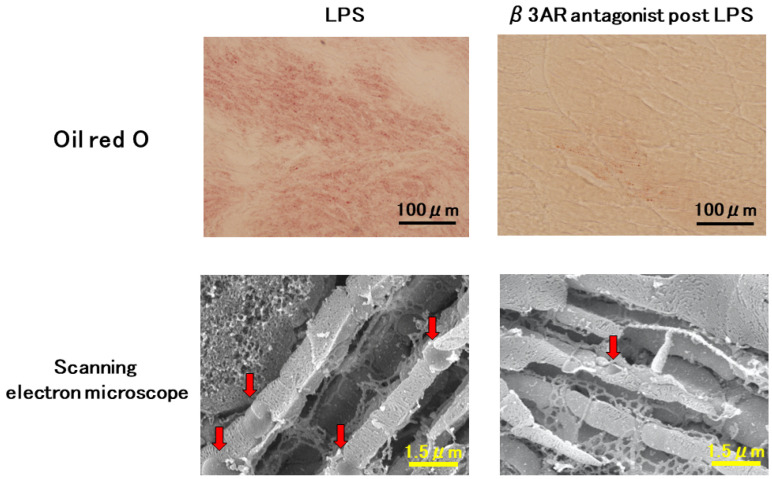
Histological analysis of septic myocardium. Oil Red O staining demonstrates the accumulation of lipid droplets (LDs) in heart tissues of LPS injected mice. β3AR antagonist significantly reduces LDs in heart tissues post LPS injection. Electron microscope analyses also exhibits a lot of LDs around mitochondria in LPS treated heart tissues. β3AR antagonist clearly attenuates accumulation of LDs. Red arrows show lipid droplets. LPS, lipopolysaccharide; β3AR, beta 3 adrenergic receptor.

**Table 1 ijms-25-04710-t001:** Noncoding RNA regulation for mitochondria in SICM.

NcRNA	Expression in Sepsis	Experimental Models	Mechanisms of Actions	Roles	Reference
LncRNA RMPP	↓	LPS treated C57BL/6 mice and cardiac muscle cells (HL-1)	Regulation of miR-1-5p/HSP 70 axis	reduction of MMP and ROS	[95]
LncRNA H19	↓	H9C2 cell line	Regulation of miR-93-5p/SORBS2 axis	Attenuation of MMP and inflammation	[96]
LncRNA MIAT	↑	LPS treated male BALB/c mice and HL-1 cells	Regulation of miR-330-5p/TRAF6/NF-κB axis	Promotion of inflammation and mitochondrial ROS	[94]
MiR-210-3p	↑	CLP treated male Sprague Dawley rats and H9C2 cells	Regulation of NDUFA4	Promotion of cardiomyocyte apoptosis and mitochondrial dysfunction	[104]
MiR-21-3p	↑	LPS treated C57BL/6/mice and human plasma	Regulation of SORBS2	Attenuation of myocardial mitochondrial ultrastructure damage and autophagy	[105]
LncRNA SOX2OT	↑	LPS treated mice and LPS treated H9C2	Regulation of SOX2	Promotion of mitochondrial dysfunction with reduction of MMP and increase in ROS	[106]
LncRNA MALAT1	↑	LPS treated H9C2 and CLP treated rats	Regulation of EZH2/USP22 axis	Attenuation of cardiac injury and Nrf2 expression	[107]

Upward arrows indicate increased expression and downward arrows indicate decreased expression.

## Data Availability

Data available in a publicly accessible repository.

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
