# Peer review of "Current Perspectives of Mitochondria in Sepsis-Induced Cardiomyopathy"

_ijms, 2024, doi:10.3390/ijms25094710_

Round 1

Reviewer 1 Report

Comments and Suggestions for Authors

Dear Sirs,

The authors of this manuscript discuss the role of mitochondria in septic cardiomyopathy, emphasizing potential mitochondria-targeted therapeutic interventions. While this topic is of high interest, there are major issues that should be addressed. Therefore, I suggest the following changes to enhance the manuscript:

1.Abstract

- Line 7-14: I would rephrase as follows:  

Despite its reversibility, prognosis varies widely among patients. Mitochondria play a key role in cellular energy production by generating adenosine triphosphate (ATP), which is vital for myocardial energy metabolism. Over recent years, mounting evidence suggests that severe sepsis not only triggers mitochondrial structural abnormalities such as apoptosis, incomplete autophagy, and mitophagy in cardiomyocytes but also compromises their function, leading to ATP depletion. This metabolic disruption is recognized as a significant contributor to SICM, yet effective treatment options remain elusive. Sepsis cannot be effectively treated with inotropic drugs in failing myocardium due to excessive inflammatory factors that blunt β-adrenergic receptors.

2.Introduction:

- I would reconstruct the introduction part as follows:

1.Please provide the definition of sepsis, followed by data regarding the role of inflammation (data in lines 40-43).

2.Evidence regarding SICM.

- In particular, the following changes should be made:

1.Lines 25-27: The sentence is wrong. Sepsis is not a severe, life-threatening organ dysfunction, but a severe condition which may be complicated by life-threatening organ dysfunction. This sentence may be changed as follows:

Sepsis is a severe condition that can be complicated by life-threatening organ dysfunction resulting from an abnormal host response to infection. However, SICM specifically affects the heart.

2.Line 38: such as?

3.Lines 44-47: While evidence of pathological changes in other organs has accumulated, little has been reported on the pathology of SICM. Some animal studies have shown that while myocardial mitochondria did not exhibit morphological abnormalities, there were noticeable alterations in the surrounding endoplasmic reticulum.

4.Line 48: ‘’could be one of the mechanisms’’ may be replaced by ‘’may be a potential mechanism’’.

5.Line 50: ‘’ SICM is considered reversible in the clinical situation’’. Maybe the authors mean in the clinical practice?

6.Lines 51-57: Please use 2 sentences instead of 1.

7.Lines 61-63: Revise as follows: ‘’ Seen from a different perspective, SICM may be regarded as a metabolic disorder. Myocardial energy metabolism encompasses a diverse array of pathways, with mitochondria playing a central role as crucial organelles involved in metabolism’’.

3.The introduction part should be followed by a ''Methodology Literature Search'' section. For the methodology section of this narrative review, the authors should outline how they conducted the literature search and review process.

4.Mitochondrial Maturation and Energy Metabolism in the Myocardium:

- Lines 68-71: Please rephrase as follows:

The mitochondria account for approximately 30% of the cell volume in cardiac myocytes, highlighting their fundamental role in cardiomyocyte function. During fetal development, mitochondria in the myocardium are relatively low in quantity, resulting in decreased metabolic capacity for fatty acid oxidation (FAO).

- Lines 76-78: 60-90% instead of 60%-90% and 10-40%, respectively. The same corrections should be made in lines 81-82, and 109.

- Lines 90-93: Please use 2 sentences instead of 1.

5.Mitochondrial Structure, Calcium Transport-induced Membrane Potential.

- Please revise the title of this section in order to be more precise.

- Line 122: correct the typographical error.

- Lines 134-138: Please revise as follows:

Calcium (Ca²) plays a crucial role in the excitation and contraction of cardiomyocytes. It is responsible for orchestrating the excitation-contraction coupling process within cardiomyocytes, where Ca² facilitates muscle contraction. Additionally, mitochondrial Ca² levels regulate the excitatory and contractile functions of cardiomyocytes. Overall, Ca² is central to maintaining the intricate balance in the excitatory-contraction relationship of cardiomyocytes.

6.Pathophysiology of SICM.

- The authors should make a comment emphasizing that LPS are exclusively found in the outer membrane of Gram-negative bacteria.

- Lines 207-211: Please use 2 sentences instead of 1.

- Lines 263 and 285: Please provide a figure legend for Figures 1 and 2. It would be easier for the readers to understand the meaning of these figures.

- In the introduction, the authors refer to ferroptosis, and pyroptosis, yet no related specific evidence may be found in this section. Therefore, the authors may incorporate evidence about the role of these cell death pathways in the pathophysiology of SICM. For example:

1.The role of NLPR3 inflammasome (PMID 37904696) and the vital role of Gasdermin D (GSDMD)-dependent pyroptosis in septic cardiomyopathy by regulating mitochondrial homeostasis (PMID 34820388).

2.Ferroptosis: PMID 38237386

7.Theurapeutic strategy.

- Line 296: I would change the title as follows: Potential Therapeutic Interventions: Future Perspectives  

- Line 366: Please correct: ‘’on the other hand’’.

- The authors may consider incorporating data regarding the role of irisin in SICM as a promising intervention. I have already suggested to add data regarding the role of Gasdermin in the Section of Pathophysiology). (PMID 35653888)

- The same approach can be applied regarding ferroptosis, with the inclusion of evidence from relevant studies such as PMID 38237386, which serves as a good example.

8.Conclusions.

- Line 483. Please revise as follows: SICM represents a significant manifestation of sepsis linked with a poor prognosis.

9.Additional comment: The authors should provide a list of abbreviations.

Comments on the Quality of English Language

In my opinion, certain parts of the manuscript exhibit significant issues with English language usage. Therefore, editing for language is necessary.

Reviewer 2 Report

Comments and Suggestions for Authors

While I like the topic, there are significant issues that need to be addressed with this review. I found that section 2 on Mitochondrial maturation and metabolism was very detailed and unnecessary. If you are going to keep this section, I suggest that you provide some figures with all of the many biochemical steps. However, I feel that reviewing the detailed biochemistry of mitochondrial function is unnecessary and makes the reader lose interest in the paper. Clearly, shortening the review to get more to the pathophysiology and treatment of cardiomyopathy would improve the paper significantly. If you are going to include this section, the first paragraph (starting on line 68) makes little sense and needs to be redone. The sentence on line 76 is repeated in the next paragraph (line 81). Again, reducing this early section would improve the paper. 

There are several editing issues - the title has the letters of "of" separated as "o f". Line 122, the "T" is separated by a line from the rest of the "The". 

In several sections, you mention the word "noncording" RNA. The correct word is "noncoding". Noncoding means that the RNA is not used for transcriptional coding into amino acids. 

I would focus the paper on the pathophysiology and treatment of SICM. Making the paper more to-the-point, and not reviewing basic biochemistry would improve readership. 

Comments on the Quality of English Language

There is a need to have grammar and editing corrections. 

Round 2

Reviewer 1 Report

Comments and Suggestions for Authors

The manuscript is now improved. All the requested corrections have been made. I recommend its publication in its current form, although some minor typographical errors should be fixed (eg. line 108 found and line 312 perspectives). 

Comments on the Quality of English Language

The quality of English language is acceptable.

Please fix minor typographical errors. 

Reviewer 2 Report

Comments and Suggestions for Authors

The authors have addressed my issues and have improved the paper.